# Local-scale phylodynamics reveal differential community impact of SARS-CoV-2 in a metropolitan US county

**Miguel I. Paredes** [1,2]*, **Amanda C. Perofsky**[3,4], **Lauren Frisbie**[5], **Louise H. Moncla**[6], **Pavitra Roychoudhury**[2,7], **Hong Xie**[7], **Shah A. Mohamed Bakhash**[7], **Kevin Kong**[7], **Isabel Arnould**[7], **Tien V. Nguyen**[7], **Seffir T. Wendm**[7], **Pooneh Hajian**[7], **Sean Ellis**[7], **Patrick C. Mathias**[7], **Alexander L. Greninger**[2,7], **Lea M. Starita**[3,8], **Chris D. Frazar**[8], **Erica Ryke**[8], **Weizhi Zhong**[3], **Luis Gamboa**[3], **Machiko Threlkeld**[8], **Jover Lee**[2], **Jeremy Stone**[3], **Evan McDermot**[3], **Melissa Truong**[8], **Jay Shendure**[3,8,9], **Hanna N. Oltean**[5], **Cécile Viboud**[4], **Helen Chu**[10], **Nicola F. Müller**[2‡], **Trevor Bedford**[1,2,3,8,9‡]

1 Department of Epidemiology, University of Washington, Seattle, Washington, United States of America, 2 Vaccine and Infectious Disease Division, Fred Hutchinson Cancer Center, Seattle, Washington, United States of America, 3 Brotman Baty Institute for Precision Medicine, University of Washington, Seattle, Washington, United States of America, 4 Fogarty International Center, National Institutes of Health, Bethesda, Maryland, United States of America, 5 Washington State Department of Health, Shoreline, Washington, United States of America, 6 The University of Pennsylvania, Department of Pathobiology, Philadelphia, Pennsylvania, United States of America, 7 Department of Laboratory Medicine and Pathology, University of Washington, Seattle, Washington, United States of America, 8 Department of Genome Sciences, University of Washington, Seattle, Washington, United States of America, 9 Howard Hughes Medical Institute, Seattle, Washington, United States of America, 10 Department of Medicine, Division of Allergy and Infectious Diseases, University of Washington, Seattle, Washington, United States of America

‡ These authors are jointly supervised on this work.
* paredesm@uw.edu

**Data Availability Statement:** Nextstrain builds, BEAST XMLS, scripts, sequence information, and data can be found at https://github.com/blab/ncov-king-county. All sequences are available on

## Abstract

SARS-CoV-2 transmission is largely driven by heterogeneous dynamics at a local scale, leaving local health departments to design interventions with limited information. We analyzed SARS-CoV-2 genomes sampled between February 2020 and March 2022 jointly with epidemiological and cell phone mobility data to investigate fine scale spatiotemporal SARS-CoV-2 transmission dynamics in King County, Washington, a diverse, metropolitan US county. We applied an approximate structured coalescent approach to model transmission within and between North King County and South King County alongside the rate of outside introductions into the county. Our phylodynamic analyses reveal that following stay-at-home orders, the epidemic trajectories of North and South King County began to diverge. We find that South King County consistently had more reported and estimated cases, COVID-19 hospitalizations, and longer persistence of local viral transmission when compared to North King County, where viral importations from outside drove a larger proportion of new cases. Using mobility and demographic data, we also find that South King County experienced a more modest and less sustained reduction in mobility following stay-at-home orders than North King County, while also bearing more socioeconomic inequities that might contribute to a disproportionate burden of SARS-CoV-2 transmission. Overall, our findings suggest a role for local-scale phylodynamics in understanding the heterogeneous transmission landscape.

GenBank and GISAID with accession numbers found in S2 Table.

**Funding:** T.B. is a Howard Hughes Medical Institute Investigator. This work is supported by NIH NIGMS (R35 GM119774) and HHMI COVID-19 Collaboration Initiative award to T.B. L.H.M. is funded by NIH grant number 4R00AI147029-04. A. C.P. is funded by Gates Ventures. Sequencing of specimens by the Brotman Baty Institute of Precision Medicine was funded by Gates Ventures (Seattle Flu Study award), Howard Hughes Medical Institute (HHMI COVID-19 Collaboration Initiative award) and the CDC (contract number 200-2021-10982). Sequencing of specimens by UW Virology was funded by Fast Grants (award #2244), the CDC (contracts 75D30121C10540 and 75D30122C13720) and WADOH (contract HED26002). The funders had no role in study design, data collection and analysis, decision to publish, or preparation of the manuscript.

**Competing interests:** ALG reported receiving contract testing from Abbott, Cepheid, Novavax, Pfizer, Janssen and Hologic and research support from Gilead and Merck, outside of the described work. HC reported consulting with Ellume, Pfizer, the Bill & Melinda Gates Foundation, Glaxo Smith Kline, and Merck. She has received research funding from Emergent Ventures, Gates Ventures, Sanofi Pasteur, the Bill & Melinda Gates Foundation, and research support and reagents from Ellume and Cepheid outside of the submitted work. The funders had no role in study design, data collection and analysis, decision to publish, or preparation of the manuscript. All other authors declare no competing interests.

## Author summary

State- or county-level data collected as part of routine surveillance often mask significant local differences in SARS-CoV-2 transmission due to their lack of granularity. This leaves local public health departments with incomplete information for resource allocation. Using King County, Washington as an example of a diverse, metropolitan US county, we leveraged genomic epidemiology to understand differences in transmission between North and South King County, two adjacent regions within the same county with stark socioeconomic differences. By combining epidemiological, mobility, and demographic data, we found that these two regions had divergent SARS-CoV-2 epidemic trajectories following the start of statewide stay-at-home orders in March 2020. Our approach also revealed important differences in the role of viral importations and persistence of local viral transmission on changing SARS-CoV-2 incidence in the background of large-scale non-pharmaceutical interventions. Our work shows that we can use genomic epidemiology to reveal differences in transmission at a local scale, which can inform equitable resource allocation at a local level to reduce the burden of infectious diseases.

## Introduction

The first confirmed SARS-CoV-2 infection in the United States was detected in Washington State (WA) on January 19, 2020. Since initial detection of the virus, genomic epidemiology has played a crucial role in identifying and estimating new introductions and community transmission in WA [1–3] and throughout the US [4,5] and has motivated rapid public health interventions. While international introductions continue to seed new viral lineages into the US, the majority of transmission is driven by infections and movement at a local scale, wherein neighboring states, regions, counties, or even zip codes can have vastly different epidemic dynamics [3,6,7].

In WA, genomic epidemiology has aided in understanding the spatiotemporal variation of the SARS-CoV-2 epidemic. At a statewide level, previous studies have examined changes in the relative frequency of variant viruses and the impact of non-pharmaceutical interventions on the estimated effective population size of the virus [2]. Phylodynamic analyses have estimated the role of introductions in promoting community spread in the state at large and revealed an asymmetrical interplay between the eastern and western regions of the state, wherein intra-state transmission accounts for more than half of the introductions into the eastern region of WA but only for less than 30% of the introductions into western WA [3].

Even a regional view fails to capture the nuance of epidemic dynamics needed to effectively curb transmission in the state because neighboring counties and even intra-county areas are affected by epidemic and demographic heterogeneity. King County, WA is a demographically diverse, metropolitan US county that has been proactive in promoting testing and vaccination throughout the SARS-CoV-2 epidemic. Despite these efforts, studies have revealed a large degree of variation in SARS-CoV-2 infection probability and hospitalization, with communities of color disproportionately impacted [8].

Previous studies have attributed differences in local case counts to unequal reductions in mobility [9,10]. When compared to a baseline average from 2019, King County, WA as a whole shows a large decrease in mobility following the implementation of stay-at-home orders in March 2020 but differences between within-county regions are salient: North King County experienced a 60% reduction in mobility compared to the 40% reduction in South King County (Fig 1A). While South King County eventually returned to baseline levels of mobility

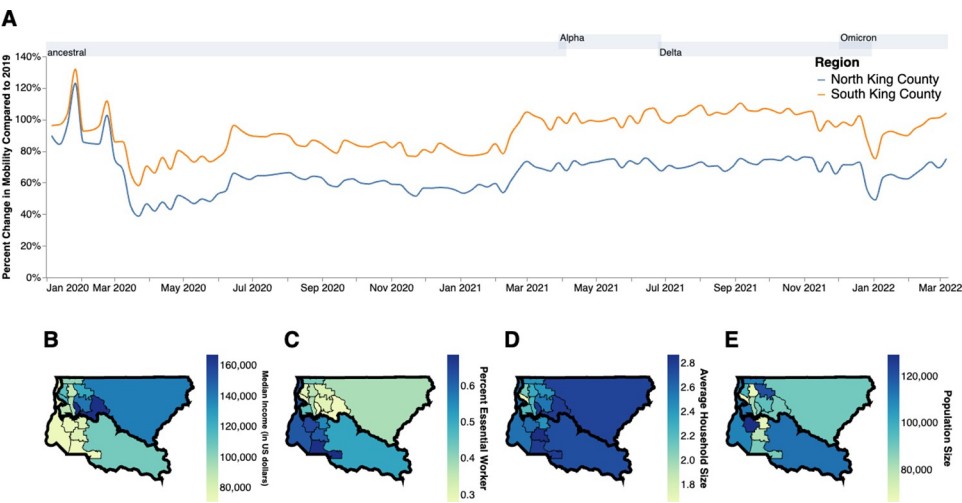

**Fig 1. Socioeconomic Characteristics of King County. A.** Percent change in mobility from Feb 2020 to March 2022 over time using average mobility in 2019 as baseline for North (blue line) and South (orange line) King County. Dashed line denotes no change compared to baseline. B,C. Median household income in 2020. (**B**) Percentage of the active workforce whose occupation is defined as "essential" from 2015–2020 (**C**) average household size from 2015–2020 (**D**) and population size (**E**) in King County by Public Use Microdata Area (PUMA). Gray shaded regions above each figure show the time periods during which ancestral virus, Alpha, Delta, and Omicron respectively represented greater than 30% of sequenced case. Geojsons for King County PUMAs were made using shapefiles from the US Census Bureau [12] and can be found here: https://github.com/seattleflu/seattle-geojson/tree/master/seattle_geojsons.

by the end of 2020, North King County was able to maintain reduced levels through March 2022. The ability to significantly reduce and maintain mobility changes has been previously attributed to socioeconomic inequities, including geographical differences in income [11] and percentage of the community that contributes as an essential worker [9]. We see a similar pattern in King County: South King County has a lower median household income, a larger percentage of essential workers in the active workforce, and a higher average household size than North King County (Fig 1B–1D), despite a smaller population size (Fig 1E).

While some studies have used genomic epidemiology to examine transmission between US counties or boroughs [5–7], here we employ phylodynamic tools to understand the fine scale spatial and temporal dynamics of SARS-CoV-2 viral transmission both within and between regions of King County, WA, as a case study of a demographically and socioeconomically diverse US metropolitan county. Using 11,602 viral sequences sampled from individuals in King County between January 2020 and March 2022, we examined the role of introductions in promoting community spread and the impact of non-pharmaceutical interventions on viral transmission dynamics.

## Results

The COVID-19 epidemic in King County, WA shows distinct spatial and temporal patterns that persisted throughout our study from February 2020 to March 2022. At the PUMA level (see Methods under Geographical scales), confirmed COVID-19 cases and hospitalizations in King County are disproportionately higher in more southern PUMAs than in northern PUMAs (Fig 2A and 2B) during almost every time period analyzed. During the last time period encompassing the BA.1 Omicron wave, from December 2021 to March 2022, we observed a more equal geographic distribution of confirmed COVID-19 cases, but COVID-19 hospitalizations continue to disproportionately affect southern regions.

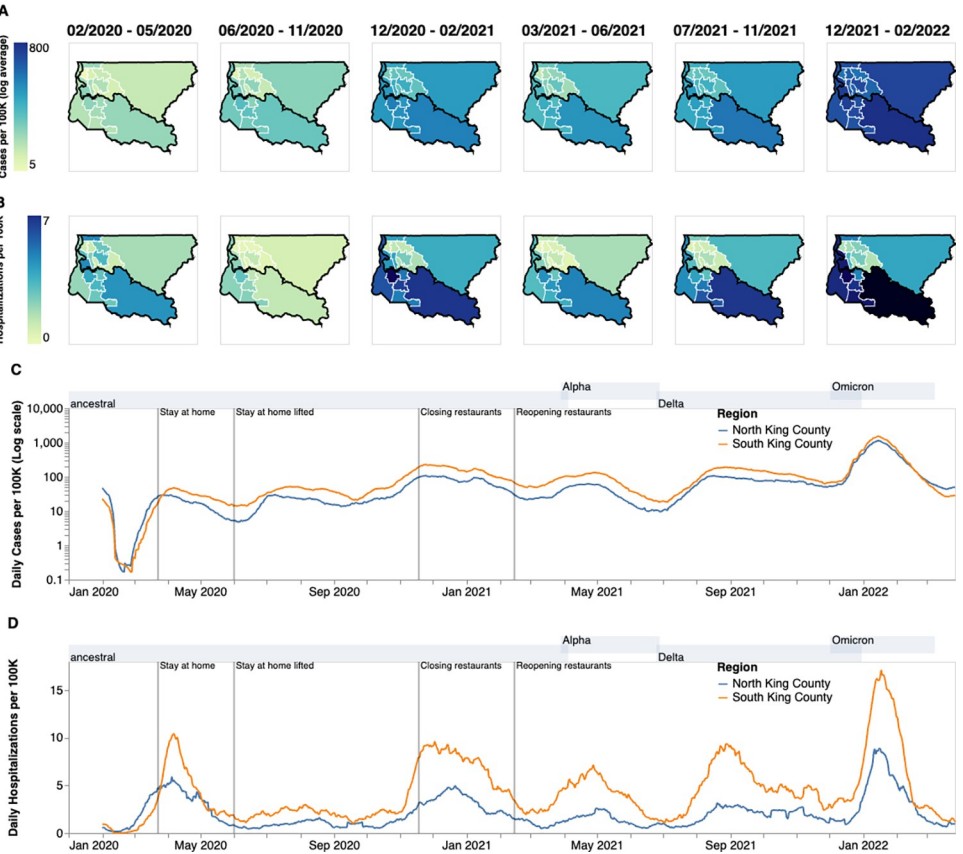

**Fig 2. Descriptive Epidemiology of SARS-CoV-2 Epidemic in King County, WA. (A, B)** Confirmed positive cases (A) and hospitalizations (B) per 100,000 individuals of SARS-CoV-2 in King County by Public Use Microdata Area (PUMA) averaged for each of the six waves of the epidemic up until March 2022. Dark borders denote geographical boundaries between North and South King County **(C, D)** Daily positive cases and hospitalizations of SARS-CoV-2 from February 2020 to March 2022 by region of King County smoothed with a 14 day rolling average. Blue denotes North King County; Orange denotes South King County. Gray shaded regions above each figure show the time periods during which ancestral virus, Alpha, Delta, and Omicron respectively represented greater than 30% of sequenced case. Geojsons for King County PUMAs were made using shapefiles from the US Census Bureau [12] and can be found here: https://github.com/seattleflu/seattle-geojson/tree/master/seattle_geojsons.

Due to the salient differences between northern and southern PUMAs, we then divided King County into two regions, North and South, and analyzed COVID-19 cases and hospitalizations continuously over time (Fig 2C and 2D). From January 2020 to the end of March 2020, during the beginning of the epidemic, we see that cases and hospitalizations are slightly higher in North King County. However, starting in April 2020 soon after a stay-at-home order on March 23, South King County consistently had higher confirmed cases and hospitalizations per capita than North King County, a trend that mostly persisted throughout the time period studied, except during the Omicron wave when cases were similar in both regions. Time series of cases and hospitalizations replicated the geographical trends seen in Fig 1A and 1B: while the difference in the number of confirmed cases seemed to contract in during the BA.1 Omicron wave (Dec 2021–March 2022), the magnitude of the difference in hospitalizations remains roughly constant, with South King County disproportionately burdened.

To investigate transmission dynamics between and within these two King County regions, we analyzed 11,602 sequenced King County viruses alongside contextual sequences from around the world. Following the creation of time-resolved phylogenies using Nextstrain [13],

we split the sequences into local outbreak clusters using parsimony-based clustering to identify groups of sequences whose ancestral states were inferred to be in King County (see Methods, S1 Fig). We identify 5964 clusters and find that the number of clusters increases over the time in both regions (Fig 3A), most likely due to an increase in the number of cases being sequenced in WA. Additionally, we find that the majority of clusters are single introductions (n = 5,095), with larger clusters increasingly rare (Fig 3B, clusters with more than 10 sequences were excluded for clarity). South King County has a greater mean cluster size (South: 1.87; North: 1.61; two-sample t-test p-value: 0.048) as well as a larger maximum cluster size (max South cluster size of 280 vs max North cluster size of 150). Fig 3C shows the phylogenetic tree of all clusters with 5 or more sequences with inferred geographic location as coloring.

We also analyzed the inferred ancestral location for all clusters over time divided out by the dominant variant waves (S2 Fig). We found that Alpha and Delta arrived first into King County mainly from other US states before spreading into the larger WA region, with Alpha also arriving from the UK where it originated. As time progressed, the source of introductions switched from mainly North America (excluding WA) to predominantly from within Washington (excluding Omicron which was introduced into King County primarily from WA). Additionally, we saw that North King County has a larger proportion of viral introductions coming from outside WA, while the majority of introductions into South King County come from within the state.

We then employed phylodynamic inference methods on the identified outbreak clusters to analyze SARS-CoV-2 spread in the county. Following subsampling, we used a MASCOT-GLM approach with relevant predictors on a random subsample of 3000 sequences from our dataset of local outbreak clusters to reconstruct SARS-CoV-2 transmission dynamics (S3 Fig). Phylodynamic estimates of the effective population size ($Ne$) of the virus in both King County regions over time mirror patterns seen in both confirmed COVID-19 hospitalizations and cases: while the $Ne$ in North King County is initially greater until the end of March 2020, following WA stay-at-home orders, we find a consistently greater $Ne$ in South King County throughout the study period (Fig 4A). We also find that hospitalizations one week in the future was the most informative predictor for effective population size in our model (Fig 4B), while the migration rate predictors were not significantly informative (Fig 4C).

We next analyzed the posterior set of phylogenies produced by the MASCOT-GLM analysis to understand viral circulation within and between the two regions. Given the higher estimated $Ne$ in South King County, we quantified the average persistence time of viral transmission chains in each region (Fig 5A, see Methods). While the average monthly persistence time remained relatively equal between the two regions during the early stages of the epidemic, following May 2020 up until 2022, we see that transmission chains in South King County consistently have significantly higher persistence times than in North King County, with the mean local transmission length averaged over the entire time period of 21.5 days in South King County and 13.5 days in North King County. We see an increase in average persistence times in both regions during large waves of COVID-19 cases attributable to the introduction of a new variant with a transmissibility advantage (such as in late 2020- early 2021 with the introduction of Alpha) and the relaxation of stay-at-home order, with South King County consistently having longer persistence times.

To understand if these longer transmission chains in South King County could be due to a higher number of viral introductions from outside the county, we reconstructed the ancestral states of each *a priori* defined King County transmission cluster to quantify the relative number of introductions into each region (Fig 5B). While greater than 50% of introductions prior to May 2020 were into South King County, the majority of the time period studied was characterized by a greater relative proportion of introductions from outside into North King County.

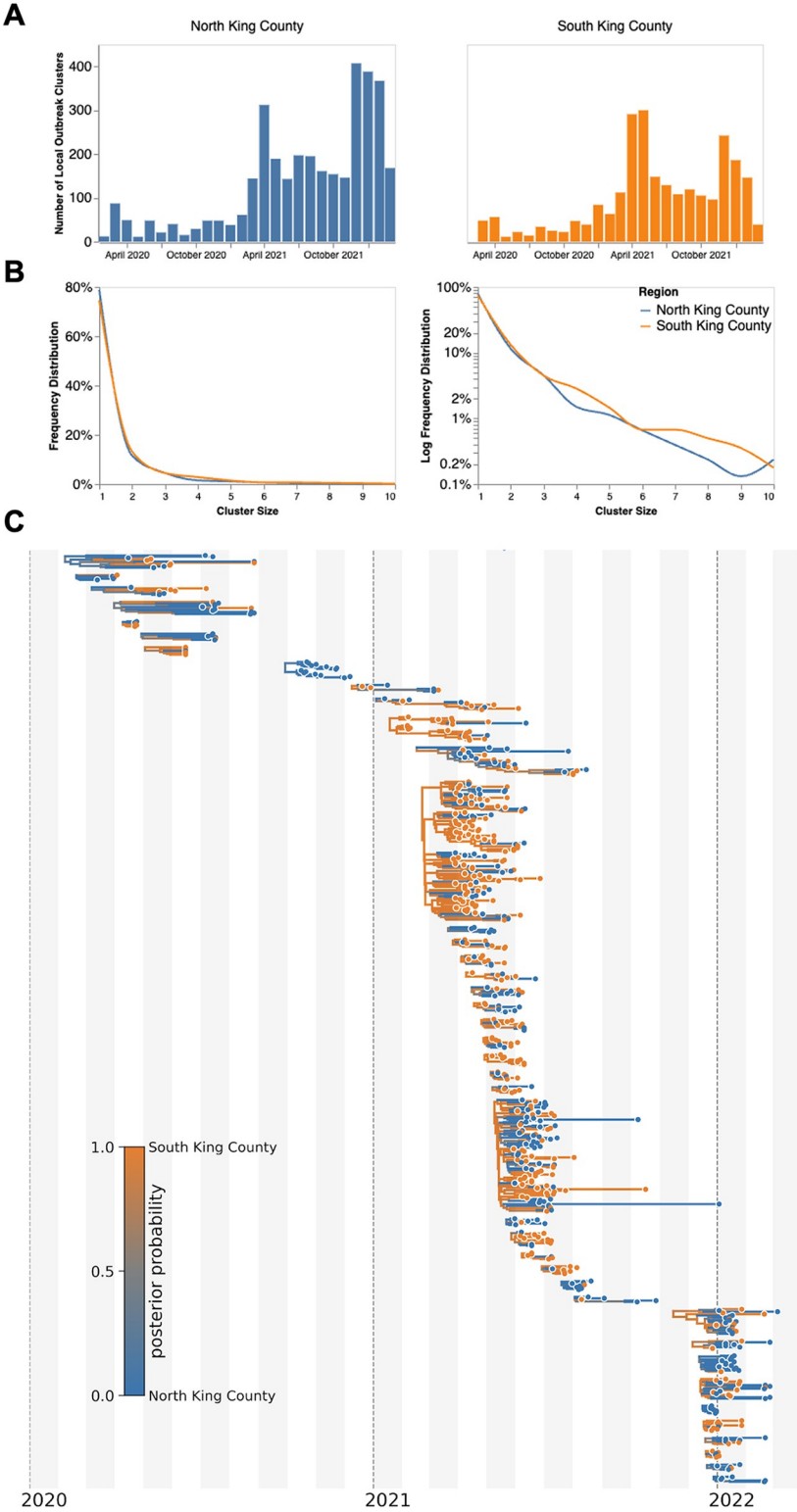

**Fig 3. Representative SARS-CoV-2 Clusters by Region in King County.** We combined more than 11,500 SARS-CoV-2 genomes from King County with more than 45,000 contextual sequences from around the world and built a time-resolved phylogeny. King County outbreak clusters were then extracted using a parsimony based clustering approach. We inferred geographic transmission history between each region using MASCOT-GLM. Here, we display the number of clusters over time by King County Region (**A**), the frequency of cluster size by region on a

linear (**B left**) and log (**B right**) scale (up to a cluster size of 10. Larger clusters exist but were excluded from the graph for clarity), and the maximum clade credibility tree of all clusters with five or more sequences (**C**) where color represents posterior probability of being in South King County. The x-axis represents the collection date (for tips) or the inferred time to the most recent common ancestor (for internal nodes). Blue denotes North King County, Orange denotes South King County.

These fine scale phylodynamic analyses also allow us to investigate the interplay between local regions. Introductions from outside regions have been shown to play a driving force in maintaining local outbreaks [14] but often these introductions are focused on interstate or international travel. Here we quantify the interplay between two inner-county regions, examining the number of transmission events that occur between North and South King County (Fig 5C). By quantifying the number of migration jumps between the two regions, we see a clear pattern emerge in which prior to June 2020 when WA lifted emergency stay at home orders, there was little difference in the number of transmission events between regions. Following the elimination of the stay-at-home orders however, transmission events become asymmetrical, where we consistently see disproportionally more transmission from South King County to North King County than in the opposite direction, with the largest differences occurring in the beginning months of 2021.

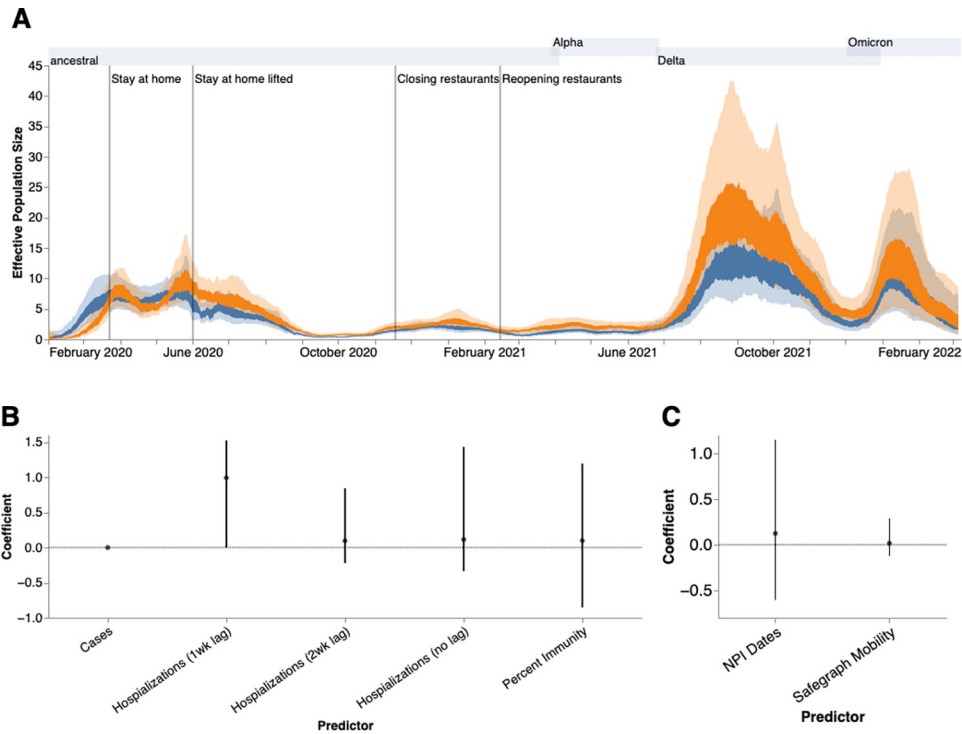

**Fig 4. Phylodynamic Analysis via MASCOT-GLM. (A)** Estimates of effective population sizes from Feb 2020 to March 2022 in North (blue) and South (orange) King County using 3000 randomly subsampled sequences. The inner band denotes the 50% highest posterior density (HPD) interva,l and the outer band denotes the 95% HPD interval. Vertical gray lines denote dates of non-pharmaceutical interventions in Washington State. **(B)** Estimates of model predictor coefficients for *Ne* estimation and **(C)** for migration rate estimation. All of the predictors displayed on the x-axis were included in the analytic model. Dark line represents median estimates, light bands represent 95% HPD. Gray shaded regions above each figure show the time periods during which ancestral virus, Alpha, Delta, and Omicron, respectively represented greater than 30% of sequenced case.

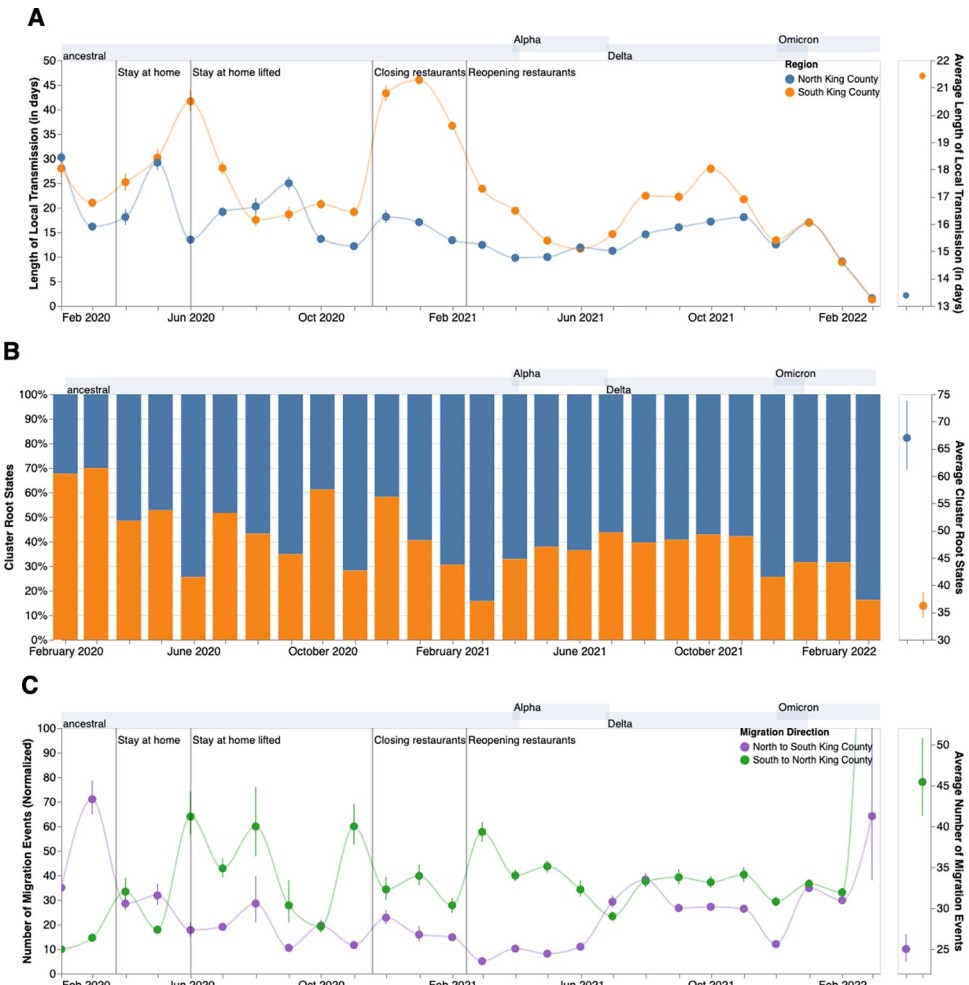

**Fig 5. Within and Inter-Regional Dynamics in King County inferred from pathogen genomes and relevant covariates. A.** Persistence time (in days) of local transmission chains over time in both regions of King County. Accompanying graph showing persistence times averaged over the entire time period for both regions with error bars denoting 95% CIs. **B.** Inferred reconstruction of ancestral state for each transmission cluster over time. Blue denotes initial introduction in North King County and orange denotes initial introduction in South King County. Average values are normalized to 100% over time. The Accompanying graph showing inferred introductions averaged over the entire time period for both regions with error bars denoting 95% CIs. **C.** Number of migration events from North to South King County (purple) and from South to North King County (green) over time. Bands denote 95% CI. The accompanying figure shows the number of migration events between the two regions averaged over the entire time period with error bars denoting 95% CIs. Gray shaded regions above each figure show the time periods during which ancestral virus, Alpha, Delta, and Omicron respectively represented greater than 30% of sequenced cases.

Given the higher number of introductions into North King County but the larger *Ne* and longer transmission chain length in South King County, we sought to estimate the relative contribution of introductions versus local community spread in driving the epidemic in both King County regions. To do so, we calculated the percentage of new cases from introductions in each region using the estimated changes in *Ne* over time as well as the estimated rates of introduction both from outside King County and from the neighboring inner-county region. We estimated a relatively higher percentage of cases due to introductions in South vs North King County prior to emergency stay-at-home order in WA on March 23, 2020 (Fig 6A). Following the stay-at-home order, the pattern switched and was largely constant throughout the epidemic, with North King County averaging about 35% of new cases from introductions

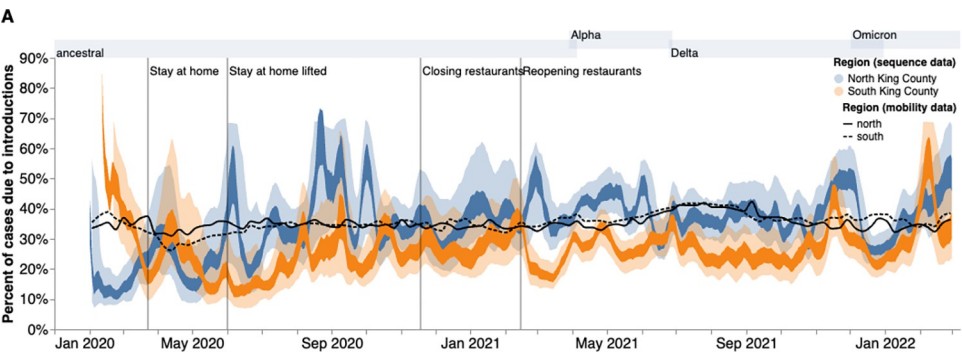

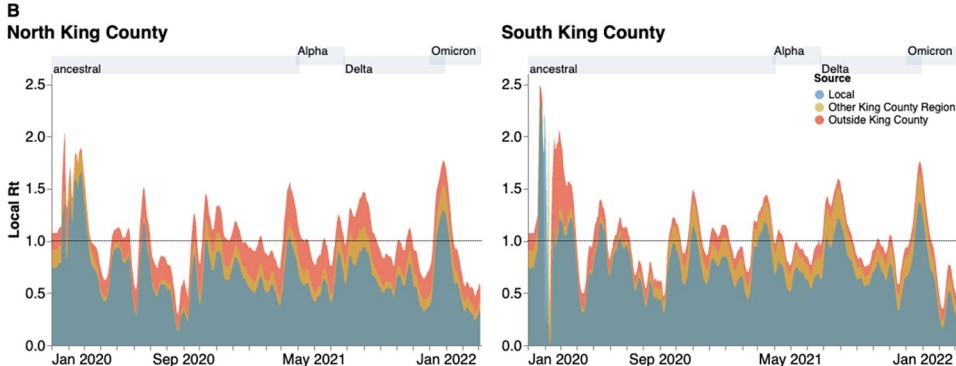

**Fig 6. Phylodynamic estimates of the differential impact of introductions and local spread on transmission dynamics of SARS-CoV-2 by region in King County. (A)** Percentages of new cases due to introductions were estimated as the relative contribution of introductions to the overall number of infections in the region. The inner area denotes the 50% HPD interval and the outer area denotes the 95% HPD interval. Blue = North King County; Orange = South King County. Black lines represent the same calculation using SafeGraph mobility data as parameter approximations. Solid black line is for North King County; Dashed black line is for South King County. **(B)** Estimates of local *Rt* highlighting the contribution of introductions from outside King County (red) and from the neighboring King County region (gold) on local transmission in each King County region. Dashed line denotes an Rt of 1. Estimates were smoothed using a 7 day rolling average. Estimates higher than 1 suggest an exponentially growing epidemic. Gray shaded regions above each figure show the time periods during which ancestral virus, Alpha, Delta, and Omicron respectively represented greater than 30% of sequenced cases.

versus local spread while only about an average of 25% of new cases were estimated to be from introductions in South King County. To further support this estimate, we calculated the percentage of visits to POIs in North and South King County for devices having an outside home location using SafeGraph mobility data. We find similar estimates ranging from about 25%-40% throughout time (Fig 6A, black lines).

To better compare transmission dynamics between the two regions, we next used the effective population size dynamics to compute *Rt*, the time-varying effective reproductive number (Figs 6B and S4). Additionally, we also employed our estimates of the percentage of new cases that are due to introductions to separate out the effects of local transmission and introductions on *Rt*. We find that the *Rt* for both regions closely follows variant waves, with an *Rt* above 1, which implies increasing transmission, matching with dates of increased case counts. Additionally, by separating out contributions into being from local transmission, introductions from the neighboring King County region, or introductions from outside King County, we find that local transmission is the main contributor to *Rt* in both regions but that introductions have a differential impact. We see that introductions as a whole play a much larger role in promoting and maintaining transmission in North King County, with outside regions being the

main contributor of introductions. In South King County, *Rt* is more driven by local within-region spread, with introductions from North King County being more influential than introductions from outside the county.

Phylodynamic estimates of epidemic dynamics were similar regardless of subsampling strategy used (S5 and S6 Figs).

## Discussion

The surge of whole genome sequencing has enabled large-scale investigation into key COVID-19 epidemiological dynamics. Yet, genomic epidemiology can also be employed to analyze transmission patterns at a local scale to aid in policy making and intervention evaluation. Here, we examined fine-scale SARS-CoV-2 transmission dynamics at a sub-county level for King County, WA, a large metropolitan area with a demographically diverse population.

We used phylodynamic methods to reconstruct the epidemic in King County from January 2020 to March 2022 and examine within-region dynamics and their interplay from pre-identified local outbreak clusters. We divide King County into North and South, informed by the clear differences in outcomes (cases and hospitalizations) at the PUMA level, in which South King County has been disproportionately affected despite having a smaller population size (673,548 in South versus 1,400,211 in North King County in 2020 [15]). We estimated that for the majority of the time period studied, introductions accounted for a larger percentage of new cases in North than in South King County (Fig 5). While a higher proportion of introductions among new cases can be attributed to either a higher rate of introduction or a lower local transmission rate, we find evidence of a greater number of viral introductions into North King County over time, from both outside and within the county, but longer chains of local transmission in South King County (Fig 5). Together, our data suggest a larger impact of introductions in North King County and a larger role of local community spread in South King County in driving the respective regional epidemics. This conclusion is supported via our *Rt* estimates, or the time-varying estimate of secondary infections, which show that outside introductions play a significant role in transmission in North King County while local spread is more contributory in South King County (Fig 6). Importantly, cases being driven by a higher percentage of introductions can be due to either an increase in introductions from outside, a decrease in local spread, or a combination of both.

Given the smaller population size in South King County, one potential explanation for higher local spread in that region is reduced access to social and economic capital and health care resources needed to curb community transmission. Previous studies looking at SARS-CoV-2 test positivity in King County at a census tract level have found that a higher test positivity was associated with various socioeconomic indicators including lower educational attainment, higher rates of poverty, and high transportation costs [16,17]. Additionally, they found that communities with a higher proportion of people of color, which are more likely to be located in South King County, were also associated with higher test positivity in 2020. Hansen et al. [17], specifically found that having a place of residence in South King County was associated with SARS-CoV-2 test positivity. The stark contrast in health outcomes between North and South King County has been previously attributed to historical redlining and systemic racism, whereby decades of racial segregation prevented communities of color from residing in northern areas of Seattle and were forced into the south into present day South King County [18,19].

The associations between test positivity and socioeconomic status are not a unique King County phenomenon; they have been found in various metropolitan areas around the US [9,10,20]. Similarly, a previous study that used phylodynamics to analyze differences in

SARS-CoV-2 spread in two Wisconsin counties found that the county with the highest basic reproductive number, an approximate measure of local spread in a naive population, was also the county with the higher proportion of people in poverty and lower access to health as well as with the highest proportion of communities of color, which mimics the transmission dynamics and demographic differences seen at a within-county level in King County [6]. While we are unable to ascribe causality, our work adds to the growing body of literature showing a correlation between geographic differences in SARS-CoV-2 transmission and socio-economic inequities potentially related to the ability to reduce mobility following non-pharmaceutical interventions.

Our results are not without limitations. Whole genome sequencing in WA is conditional on laboratory-confirmed testing in which sample quality must meet minimum requirements in terms of PCR cycle threshold, potentially biasing our dataset towards more symptomatic cases, although previous studies have found no significant difference in viral load between symptomatic and asymptomatic individuals [21–23]. Additionally, the changing availability of genomic sequencing, as well as of at-home testing, is impacting the chance a case shows up in our data through the period studied (see Fig 4B). In order to limit the impact of the increased use of at-home antigen testing, we limited our analysis to only include sequences from before April 2022. Multiple subsampling strategies were considered and implemented in an effort to account for this variation (S5 and S6 Figs).

Our phylodynamic analyses are conditioned on inferred King County sequence clusters that are found through the incorporation of contextual sequences from around the world into a temporally-resolved phylogeny. As such, it is possible that differential sampling from other locations could impact our identified clusters. Limited SARS-CoV-2 sequence diversity, especially during periods of rapid transmission, could impact our ability to break up larger clusters [24], which might lead to collapsing multiple introductions into King County into shared clusters. Prior studies have used GLM approaches to ameliorate this bias [25], similar to our use of MASCOT-GLM. Optimally, we would like to avoid having to a priori define local outbreak clusters entirely by, for example, explicitly accounting for locations outside of King County in the model. This is currently not possible due to the additional computational cost of explicitly considering an outside deme. Additionally, Bayesian coalescent models assume random sampling of infected individuals, meaning that targeted sampling, such as super spreader events or contact tracing, could bias our phylodynamic estimations. Such sampling from outbreak analyses may also not be constant through time, complicating *Ne* inferences. Lastly, our *Rt* calculations assume that the change in *Ne* over time is proportional to the change in the number of infected individuals over time.

The transmission dynamics of the SARS-CoV-2 pandemic have been highly heterogeneous across countries. Here we show that even different areas of the same metropolitan region can have different trajectories. Changes in incidence throughout the course of an epidemic can be driven by changes in local transmission, importations, or both. Common methods to estimate incidence and changes in incidence via Rt often ignore or are unable to quantify these differences [26, 27], leading to situations where local health departments have limited information with which to tackle growing case counts. Our local scale genomic epidemiology approach can reveal these differences by quantifying the contribution of importations and local transmission on Rt (Fig 6B) through the joint integration of genomic and epidemiological information. Quantifying changes and differences in contribution to incidence can directly lead to tailored interventions. For example, in an area where incidence is driven mostly by outside viral introductions, interventions could focus on limiting their impact by implementing testing at the airport or quarantine for recent travelers. Meanwhile, ramping up testing, vaccination, and masking as well as providing medical and economic aid to promote quarantine and isolation

without furthering income inequities could be more impactful for areas where local community transmission is the main driver of epidemic growth.

## Methods

### Ethics statement

The Washington State Institutional Review Board designated this study as exempt. Sequencing and analysis of samples from the Seattle Flu Study was approved by the Institutional Review Board (IRB) at the University of Washington (protocol STUDY00006181). Sequencing of remnant clinical specimens at UW Virology Lab was approved by the University of Washington Institutional Review Board (protocol STUDY00000408).

### Experimental design and data sources

For this retrospective phylodynamic study, we aimed to understand local SARS-CoV-2 transmission dynamics in a diverse, metropolitan county. We analyzed 11,602 whole genome SARS-CoV-2 sequences from King County, WA and 69,588 genome sequences from around the world downloaded from GISAID [28] with sample collection dates between February 1 2020 and March 6 2022. In order to analyze local scale phylodynamics, ZIP code information for our primary dataset from King County was obtained from the Washington State Department of Health (WADOH) on March 22, 2022. 7289 (62%) of genomes from King County were sequenced by UW Virology and 2631 (22%) of genomes from King County were sequenced by Seattle Flu Study / Brotman Baty Institute for Precision Medicine. Three other laboratories (Altius, CDC and WA PHL) sequenced the remaining 1,917 (16%) of genomes collectively.

Time series of zip code-aggregated cases and hospitalizations were found on WADOH and Public Health Seattle King County's (PHSKC) Covid Data Dashboard [29]. Publicly available demographic information by ZIP code was obtained through the U.S. Census Bureau's American Community Survey (ACS). This study utilized both ACS 2015–2019 (5-Year Estimates) and ACS 2020 [15].

Additionally, we obtained mobile device location data from SafeGraph (https://safegraph. com/), a data company that aggregates anonymized location data from 40 million devices, or approximately 10% of the United States population, to measure foot traffic to over 6 million physical places (points of interest) in the US [30]. We estimated population mobility within and between North and South King County and the in-flow of visitors residing outside of King County from January 2019 to March 2022, using SafeGraph's "Weekly Patterns" dataset, which provides weekly counts for the total number of unique devices visiting a point of interest (POI) from a particular home location. Points of interest (POIs) are fixed locations, such as businesses or attractions. A "visit" indicates that a device entered a building or the spatial perimeter designated as a POI. A "home location" of a device is defined as its common night-time (18,00–7,00) census block group (CBG) for the past 6 consecutive weeks.

### Geographic scales

To understand local-scale dynamics, most of this study was focused on geographic scales finer than the county level. We divided King County into both Public Use Microdata Areas (PUMAs), which are non-overlapping, statistical geographic areas containing no fewer than 100,000 people each, and general regions, North and South. Information as to how we aggregate ZIP codes into PUMAs and PUMAs into North and South can be found in S1 Table.

## Maximum likelihood tree generation

A temporally-resolve phylogeny was created using the Nextstrain [13] SARS-CoV-2 workflow (https://github.com/nextstrain/ncov), which aligns sequences against the Wuhan Hu-1 reference using nextalign (https://github.com/nextstrain/nextclade), infers a maximum-likelihood phylogeny using IQ-TREE [31] with a GTR nucleotide substitution model, and estimates molecular clock branch lengths using TreeTime [32]. All sequences were downloaded from the GISAID EpiCoV database on May 26 2022 [28].

In order to capture the SARS-CoV-2 epidemic in King County with high resolution and computational efficiency, we created four separate temporally resolved phylogenies that span from February 2020 to March 2022. To do so, we created specific phylogenies for Omicron (Nextstrain clades 21K, 21L, 21M comprising 2856 King County Sequences and 18,817 contextual sequences from around the world), Delta (Nextstrain clades 21A, 21I, 21J comprising 2955 King County Sequences and 19,197 contextual sequences from around the world), Alpha (Nextstrain clade 20I comprising 2941 King County Sequences and 15,406 contextual sequences from around the world), and all other SARS-CoV-2 lineages (2850 King County Sequences, 16,168 contextual sequences from around the world). These builds provided higher resolution during epidemic waves while also being mutually exclusive to sequences found in the alternative builds.

Contextual sequences are needed in order to investigate how King County samples relate to regional and global viral diversity, and to identify local outbreak clusters specific to King County. Given that cluster identification is conditional on the number of background sequences that interdigitate large clades on the phylogeny, we attempted to maximize the number of contextual sequences within the bounds of reasonable computational efficiency. We prioritized sequences from WA and North America in order to optimize regional context. For each variant, we specified contextual data sampling to include up to 10,000 genomes per time-period from WA, sampled from all counties and months, up to 7000 genomes per month from other US states, and up to 5000 genomes per month from the rest of the world. In each variant-specific phylogeny, contextual sequences comprise 83–86% of the total number of sequences. While we expect the number of the clusters to increase with an increasing number of contextual sequences, prior work has shown that changes in the proportion of background sequences that make up the analytical dataset above a proportion of 50% have a limited impact on the number of clusters identified and mean cluster size (S13 Fig in [2]), and downstream phylodynamic analyses.

Phylogeographic reconstruction of spread around King County was conducted using the same Nextstrain workflow via ancestral trait reconstruction of PUMAs and North and South region geographic attributes. Metadata on ZIP code, PUMA, and region was manually added to the GISAID metadata using the ZIP code information obtained from WADOH as described above.

## Clustering

To identify local outbreak groups in King County, we clustered all King County sequences based on inferred internal node location. Following Müller et al [2], we used a parsimony-based approach to reconstruct the locations of internal nodes. Briefly, using the Fitch parsimony algorithm, we inferred internal node locations by considering only two sequence locations: King County and then anywhere else. We then identified local outbreak clusters by selecting groups of sequences in which all their ancestral nodes were inferred to be from King County, up until there was a change in location.

After identifying relevant King County clusters from each of the four variant Nextstrain builds, we then annotated the clusters in a combined dataset.

## Subsampling

To reduce computation times in subsequent MCMC analyses, we utilized three different subsampling schemes. Three thousand sequences from King County, WA from identified clusters were chosen either at random, through equal temporal subsampling for every year-week in the studied time period, or via weighted subsampling informed by daily hospitalization counts smoothed using a 14-day rolling average. The random subsampling scheme with 3000 sequences was chosen for the main result as it allowed for better resolution during variant waves.

## MASCOT GLM on multiple local outbreak clusters

To analyze the transmission dynamics within and between South and North King County, we used an adapted version of MASCOT [33] on the 3000 subsampled King County clusters and sequences. MASCOT is an approximate structured coalescent approach [34] that models how lineages coalesce (share a common ancestor) within the same locations or migrate between them. In order to distinguish between local transmission and transmission occurring outside of King County, we extended MASCOT to jointly infer coalescent and migration rates from local outbreak clusters [2]. In short, we model the transmission dynamics in King County as a structured coalescent model. We then model the introduction of lineages into King County (independent of whether it is North or South King County) as a backwards in time process of lineages having originated from outside King County. This backwards in time process is assumed to be independent of the transmission dynamics in King County and occurs at a rate given by the introduction rate [2]. The rate of introduction that is estimated as part of the MCMC is allowed to vary over time.

We used generalized log-linear models [35] to estimate whether COVID-19 hospitalizations, cases, seroprevalence, NPIs, and mobility are predictive of SARS-CoV-2 effective population sizes and migration rates over time. The model included error terms to account for observation noise and omitted predictor variables. We implemented a MASCOT-GLM [35] analysis on King County transmission clusters with BEAST2 [36] software, allowing the effective population sizes and the rates of introduction to change every day and every 14 days, respectively. We performed effective population size and migration rate inference using an adaptive multivariate Gaussian operator [37] and ran the analyses using an adaptive Metropolis-coupled MCMC [38].

## Empirical predictors

We chose several predictors to inform estimates of the migration and effective population size of SARS-CoV-2 in King County regions. To inform the effective population size, we used daily COVID hospitalizations (lagged 1–3 weeks), daily confirmed SARS-CoV-2 cases, and percent immunity against SARS-CoV-2 in Western Washington.

Percent immunity for Western Washington was found via the Nationwide COVID-19 Infection- and Vaccination-Induced Antibody Seroprevalence from the Centers for Disease Control (CDC) [39]. To include daily values, the monthly seroprevalence surveys estimates were plotted, fit to a spline and daily percent immunity values based on the fitted spline were extrapolated for the time period studied to include as a predictor.

We also used dates of non-pharmaceutical interventions (NPIs) in WA and between-region mobility to inform migration rates between North and South King County. Dates of NPIs

were found as part of the COVID-19 US State Policy Database [40]. NPIs included are start and end of emergency stay at home orders as well as closing and reopening of bars and restaurants. We chose not to include the opening and closing of public schools due to a high degree of overlap with the NPIs already included. Washington State closed down public schools on March 16th, 2020, which was only a week before the statewide shelter in place was issued on March 23rd, 2020. Similarly, public schools returned to in-person instruction on April 5th, 2021, which is near to the date of restaurant reopening at the end of February 2021.

To measure movement between North and South King County, we extracted the home CBG of devices visiting either North or South points of interest (POIs) and limited our dataset to devices with home locations in South King County visiting North King County POIs, or vice versa, and to POIs that had been recorded in SafeGraph's dataset since January 2019. For each POI in each week, we excluded home census block groups with fewer than five visitors to that POI. To adjust for variation in SafeGraph's panel size over time, we divided Washington's census population size by the number of devices in SafeGraph's panel with home locations in Washington state each month and multiplied the number of weekly visitors by that value. To estimate the total number of *visits* from each home CBG each week, we multiplied the number of weekly visitors by the total number of visits divided by the total number of unique visitors in Washington state each week. For each direction of movement, we summed these adjusted weekly visits across POIs and measured the percent change in movement from North to South or South to North over time relative to the average movement observed in all of 2019.

## Posterior processing

Parameter traces were visually evaluated for convergence using Tracer (v1.7.1) [41], and 10% burn-in was applied for all phylodynamic analyses. All tree plotting was performed with baltic (https://github.com/evogytis/baltic) and data visualizations were done using Altair [42]. We summarized trees as maximum clade credibility trees using TreeAnnotator and visually inspected posterior tree distributions using IcyTree [43].

Transmission between regions was calculated by measuring the number of migration jumps from North to South King County and vice versa walking from tips to root in the posterior set of trees. In order to account for unequal sampling between the two regions, the rate of migration was estimated as the total number of migration jumps per month in each region divided by the average branch lengths for that region for the same month.

Persistence time was measured by calculating the average number of days for a tip to leave its sampled location (North vs South), walking backwards up the phylogeny from the tip up until node location was different from tip location (following Bedford et al. [44]).

## Estimating percentage of new cases due to introductions

We estimated the percentage of new cases due to introductions for both North and South King County by adapting the methods previously described in Müller et al [2]. The percentage of cases due to introductions $\pi$ at time $t$ can be calculated by dividing the number of introductions at time $t$ by the total number of new cases at time $t$. We first represented the total number of new cases in a region as the sum of the number of introductions and the number of new local infections due to local transmission, resulting in the following equation:

$$\pi(t) = \frac{\text{of introductions}(t)}{\text{of new local cases}(t) + \text{of introductions}(t)}$$

We estimated the number of new local cases at time $t$ by assuming the local epidemic in each King County region follows a simple transmission model, in which we estimate the

number of new cases at time $t$ as the product of the transmission rate $\beta$ (new infections per day per individual) multiplied by the number of people already infected in that region $I$. For the number of introductions, we similarly assumed that the number of introductions equals the product of the rate of introduction (introductions per day, which we refer to as migration rate $m$) and the number of people already infected in that region $I$. We use the number of infected individuals in the destination region rather than the origin region for calculating the number of introductions since the approximate structured coalescent approach models epidemic processes as backwards-in-time, resulting in the equation containing only information about the number of infected individuals in the destination region. We then rewrote the above equation as

$$\pi(t) = \frac{m(t)I(t)}{\beta(t)I(t) + m(t)I(t)},$$

where $I(t)$ denotes the number of infected people in that region at time $t$. Given the presence of $I(t)$ in every element, we factored out $I(t)$ to arrive at

$$\pi(t) = \frac{m(t)}{\beta(t) + m(t)}.$$

For each region in King County, we considered introductions at time $t$ to be the sum of the introductions coming into the region from outside of King County and introductions coming from the neighboring King County region. Splitting up the introductions by source of contribution, we ultimately defined the percentage of new cases due to introductions $\pi$ at time $t$ for region $y$ as

$$\pi_y(t) = \frac{m^b{}_{zy}(t) + m_{out}(t)}{\beta_y(t) + m^b{}_{zy}(t) + m_{out}(t)},$$

where $m^b{}_{zy}$ denotes the backwards migration rate per day from the neighboring King County region $z$ into region $y$, and $m_{out}$ refers to the migration rate per day into region $y$ from outside of King County.

In a transmission modeling framework, the transmission rate $\beta$ is equal to the sum of the growth rate $r$ and the per-day uninfectious rate $\delta$ where

$$\beta = r + \delta$$

To compute the growth rate in region $y$, we assume that differences in effective population size between adjacent time intervals can approximate the growth rate $r$ and thus $\frac{d(log(Ne_y))}{dt} \approx r$. In addition, we assumed that $dN_e/dt$ is independent from the rate of introduction. We calculated the growth rate of the effective population size $\frac{dNe}{dt}$ as

$$\frac{d(log(Ne))}{dt} = \frac{log(Ne(t + \Delta t)) - log(Ne(t))}{\Delta t},$$

where $Ne(t)$ denotes the effective population size of a region at time $t$. We ran our MASCOT-GLM analysis using daily time intervals but calculated $Ne$ using a rolling weekly average in order to smooth our estimates.

By also assuming an expected time until becoming uninfectious for each individual of 7 days [45], we calculated the transmission rate $\beta$ at time $t$ in region $y$ as

$$\beta_y(t) = \frac{d(log(Ne))}{dt} + \delta$$

The rate of introduction per day from outside of King County $m_{out}(t)$ into a King County region $y$ is a parameter that was directly inferred by MASCOT-GLM for each daily time interval by modeling everything outside of King County as a separate third deme.

To compute the backwards migration rate, we first calculate the forward-in-time varying migration rate $m^f_{yz}(t)$ for region $y$ into region $z$ over a linear combination of $c$ different predictors:

$$m^f_{yz}(t) = b \, exp(\sum_{i=1}^{c} w^i \sigma^i p^i(t) + e)$$

where the forward migration rate $m^f(t)$ is computed via MASCOT-GLM coefficients $w^i$, indicators $\sigma^i$, log-standardized predictor values $p^i$ for predictor $i$ and the respective error parameter $e$. The variable $b$ outside the summation refers to the overall migration rate scaler while, $w^i$ refers to the migration rate scalar for each of the individual $c$ predictors.

From the forward-in-time migration rate $m^f_{yz}(t)$, we can then calculate the backwards-in-time migration rate from state $z$ to state $y$, $m^b_{zy}(t)$, as the product of the ratio of effective population sizes $\frac{Ne_y(t)}{Ne_z(t)}$ and the calculated forward migration rates:

$$m^b_{zy}(t) = \frac{Ne_y(t)}{Ne_z(t)} m^f_{yz}(t),$$

Where $Ne_y(t)$ refers to the effective population size in region y at time $t$ and $Ne_z(t)$ refers to the effective population size in the neighboring King County region $z$ at time $t$.

In addition to the calculation of percentage of new cases due to introductions, we repeated the above calculation using only SafeGraph mobility data. We used the in-flow of visitors from outside of King County and movement between each region of King County as approximations for the number of introductions and within-region mobility as an approximation for the transmission rate, following the same equation presented above. When estimating in-flows from outside King County and within-region movement, we applied the same filtering and normalization methods used when estimating between-region movement.

## Estimating the effective reproductive number Rt

We calculated the effective reproductive number $Rt$, the time-varying average of secondary infections, in both regions, using both the daily time-varying transmission rate $\beta$ and the becoming uninfectious rate $\delta$ where $Rt = \frac{\beta}{\delta}$. Additionally, we sought to separate out the contributions of introductions versus local transmission to the $Rt$ of each region. To do so, we modified the $Rt$ equation to include the percent of new cases from introductions as an estimate of local community spread only: $Rt = \frac{\beta(1-\pi)}{\delta}$, where $\pi$ refers to the percentage of new cases due to introductions as described above.

To estimate the contribution of introductions from outside of King County separately from that of the neighboring King County region, we calculated $Rt$ using the above equation and the percent of cases from introductions as previously described but omitting introductions from

outside King County. Briefly:

$$\pi_{yz}(t) = \frac{m_{yz}(t)}{\beta(t) + m_{yz}(t)},$$

where $\pi_{yz}(t)$ refers to the percentage of cases in region $z$ due to introductions from region $y$ into region $z$ at time $t$, and $m_{yz}$ refers to the per-day migration rate from region $y$ to $z$ as derived above.

## Supporting information

**S1 Fig. Time-resolved maximum likelihood phylogenies for King County, WA by dominant variant wave with sample collection dates between February 1 2020 and March 6 2022.** Trees are filtered to highlight genomes from King County among contextual sequences from around the globe. Tip color represents the region within King County, with pink corresponding to North King County and blue representing South King County. Branches are colored based on inferred ancestry. Panel **A** represents all variant clades excluding Alpha, Delta, and Omicron (the full tree can be explored interactively at https://nextstrain.org/groups/blab/ncov-king-county/other), the other panels represent Alpha (**B**, https://nextstrain.org/groups/blab/ncov-king-county/alpha), Delta (**C**, https://nextstrain.org/groups/blab/ncov-king-county/delta), and Omicron (**D**, https://nextstrain.org/groups/blab/ncov-king-county/omicron.
(TIF)

**S2 Fig. Source of introduction for each identified King County cluster.** The left column is introductions into North King County, the right into South King County. The panels show how the inferred geographical source of each introduction changes over time as a percentage of all introductions into the regions for that time period. The top row contains all the introductions among the four different time-resolved phylogenies. Each subsequent row represents a different variant studied and is labeled accordingly.
(TIF)

**S3 Fig.** Number of local outbreak clusters over time by subsampling scheme: random (A, Blue), equal temporal weighting by year-week (B, Gold), and subsampling weighted by daily hospitalizations calculated using a 14 day moving average (C, Red).
(TIF)

**S4 Fig. $R_t$ estimation using phylodynamic estimates (Blue North King County; Orange = South King County) and case data (Black lines, solid = North King County, dashed = South King County) The inner area denotes the 50% HPD interval and the outer area denotes the 95% HPD interval.**
(TIF)

**S5 Fig. Phylodynamic estimates of SARS-CoV-2 transmission in King County with equal temporal subsampling.** Results presented above were inferred using 3000 sequences subsampled using equal temporal weighting by year-week. Analyses presented, as defined previously, are: effective population size over time (A), percent of cases due to introductions (B), and local Rt estimations divided by region and source of contribution (C). Orange denotes South King County; blue denotes North King County.
(TIF)

**S6 Fig. Phylodynamic estimates of SARS-CoV-2 transmission in King County with subsampling weighted by hospitalizations.** Results presented above were inferred using 3000

sequences subsampled using weighting by hospitalizations over time using a 14 day rolling average. Analyses presented, as defined previously, are: effective population size over time (A), percent of cases due to introductions (B), and local Rt estimations divided by region and source of contribution (C). Orange denotes South King County; blue denotes North King County.
(TIF)

**S1 Table. Geocoding for different geographical scales in King County, WA.**
(XLSX)

**S2 Table. Sequence Accession IDs and acknowledgements table.**
(CSV)

## Acknowledgments

We would like to thank Mike Famulare for assembling the geojsons of King County PUMAs from the US Census Bureau that were used in this study. Clinical and sentinel laboratories who forwarded specimens for sequencing, and sequencing laboratories that reported data to WADOH. We gratefully acknowledge all data contributors, ie the Authors and their Originating laboratories responsible for obtaining the specimens, and their Submitting laboratories for generating the genetic sequence and metadata and sharing via the GISAID Initiative, on which this research is based. We have included it in S2 Table. The WADOH Data Science Support Unit for integrating sequencing data with epidemiologic case data. We also thank SafeGraph for providing foot traffic data.

## Author Contributions

**Conceptualization:** Miguel I. Paredes, Louise H. Moncla, Nicola F. Müller, Trevor Bedford.

**Data curation:** Miguel I. Paredes, Amanda C. Perofsky, Lauren Frisbie, Pavitra Roychoudhury, Hong Xie, Shah A. Mohamed Bakhash, Kevin Kong, Isabel Arnould, Tien V. Nguyen, Seffir T. Wendm, Pooneh Hajian, Sean Ellis, Patrick C. Mathias, Alexander L. Greninger, Lea M. Starita, Chris D. Frazar, Erica Ryke, Weizhi Zhong, Luis Gamboa, Machiko Threlkeld, Jover Lee, Jeremy Stone, Evan McDermot, Melissa Truong, Jay Shendure, Hanna N. Oltean, Cécile Viboud, Helen Chu.

**Formal analysis:** Miguel I. Paredes, Amanda C. Perofsky, Nicola F. Müller.

**Funding acquisition:** Pavitra Roychoudhury, Patrick C. Mathias, Alexander L. Greninger, Lea M. Starita, Chris D. Frazar, Jay Shendure, Cécile Viboud, Helen Chu, Trevor Bedford.

**Investigation:** Miguel I. Paredes, Amanda C. Perofsky, Nicola F. Müller, Trevor Bedford.

**Methodology:** Miguel I. Paredes, Amanda C. Perofsky, Nicola F. Müller, Trevor Bedford.

**Resources:** Louise H. Moncla, Trevor Bedford.

**Software:** Jover Lee, Nicola F. Müller.

**Supervision:** Louise H. Moncla, Lea M. Starita, Hanna N. Oltean, Cécile Viboud, Helen Chu, Nicola F. Müller, Trevor Bedford.

**Validation:** Miguel I. Paredes.

**Visualization:** Miguel I. Paredes, Amanda C. Perofsky, Nicola F. Müller.

**Writing – original draft:** Miguel I. Paredes, Amanda C. Perofsky, Nicola F. Müller, Trevor Bedford.

**Writing – review & editing:** Miguel I. Paredes, Amanda C. Perofsky, Lauren Frisbie, Louise H. Moncla, Pavitra Roychoudhury, Hong Xie, Shah A. Mohamed Bakhash, Kevin Kong, Isabel Arnould, Tien V. Nguyen, Seffir T. Wendm, Pooneh Hajian, Sean Ellis, Patrick C. Mathias, Alexander L. Greninger, Lea M. Starita, Chris D. Frazar, Erica Ryke, Weizhi Zhong, Luis Gamboa, Machiko Threlkeld, Jover Lee, Jeremy Stone, Evan McDermot, Melissa Truong, Jay Shendure, Hanna N. Oltean, Cécile Viboud, Helen Chu, Nicola F. Müller, Trevor Bedford.

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
