## [Decision Letter · Decision Letter 0]

22 Dec 2023

Dear Mr. Paredes,

Thank you very much for submitting your manuscript "Local-Scale phylodynamics reveal differential community impact of SARS-CoV-2 in metropolitan US county" for consideration at PLOS Pathogens. As with all papers reviewed by the journal, your manuscript was reviewed by members of the editorial board and by several independent reviewers. The reviewers appreciated the attention to an important topic. Based on the reviews, we are likely to accept this manuscript for publication, providing that you modify the manuscript according to the review recommendations.

Both reviewers agree this is an excellent high quality paper and have a series of important small suggestions. Of note esp. is comment 6 of reviewer 2 - it is important to contextualize this paper for broad readership, and give examples of how this study can inform us beyond the particular case of north/south king county (which may very well reflect similarity to many other large cities?).

Sincerely,

Adi Stern

Academic Editor

PLOS Pathogens

Alexander Gorbalenya

Section Editor

PLOS Pathogens

Kasturi Haldar

Editor-in-Chief

PLOS Pathogens

orcid.org/0000-0001-5065-158X

Michael Malim

Editor-in-Chief

PLOS Pathogens

orcid.org/0000-0002-7699-2064

Both reviewers agree this is an excellent high quality paper and have a series of important small suggestions. Of note esp. is comment 6 of reviewer 2 - it is important to contextualize this paper for broad readership, and give examples of how this study can inform us beyond the particular case of north/south king county (which may very well reflect similarity to many other large cities?).

Reviewer Comments (if any, and for reference):

Reviewer's Responses to Questions

**Part I - Summary**

Reviewer #1: The study investigates the transmission dynamics of SARS-CoV-2 at a local level in King County, Washington, utilizing genomic, epidemiological, and mobility data from February 2020 to March 2022. The authors employ an advanced analysis method to uncover significant disparities in transmission patterns between North and South King County following the implementation of stay-at-home orders. Notably, South King County experienced higher reported cases, hospitalizations, and sustained local transmission compared to North King County, where new cases are primarily influenced by importation from outside the county. These disparities are attributed to diverse mobility and socioeconomic challenges faced by residents of South King County, emphasizing the importance of considering local conditions in pandemic management. The study is exceptionally well-written report, the results are well presented, and includes an impressive phylodynamic analysis pipeline. A few remarks are listed below for consideration.

Reviewer #2: This study uses 11,737 SARS-Cov-2 sequences sampled from January 2020 – March 2022 from people living in King County, Washington (which includes the city of Seattle) to study patterns of virus transmission, including community spread versus the role of outside introductions in driving the epidemic. These approaches have been successfully used around the world to evaluate the impact of non-pharmaceutical interventions on controlling virus transmission.

**Part II – Major Issues: Key Experiments Required for Acceptance**

Reviewer #1: - In the Methods section, please provide a comprehensive explanation of the selection process for contextual sequences, as identifying local clusters through phylogenetic inference is a critical aspect of this study.

Reviewer #2: 1. This paper discusses the role of outside introductions. Did North King county and South King county appear to have the same geographical sources for these introductions? Did the geographical source of outside introductions shift over the course of the pandemic?

2. One question during the pandemic was why alpha did not take off in the US the way it did in the UK where it emerged. When was the variant introduced into King County? Did you observe lots of introductions but not much community transmission? Any idea why?

3. Is there any parameter that is a good predictor of whether a transmission chain persists for a long period? Seasonality? Population immunity? Location of origin? Timing related to non-pharmaceutical interventions? Or does it appear to be stochastic?

**Part III – Minor Issues: Editorial and Data Presentation Modifications**

Reviewer #1: - The statement “Figure 2c shows all clusters greater than size five with respective posterior support for inferred ancestral states.” is redutant as it repeats the earlier statement.

- Please label control measures in Fig. 1 C and D, and Fig. 5A

- Typo? mzy -> myz

- In the sentence “21,976 genome sequences from around the world downloaded from GISAID”, the contextual sequences for Omicron, Delta, Alpha and other lineages exceed this total number.

Reviewer #2: 1. What is most interesting about this study is the comparison of transmission dynamics between North King County and South King County and how their differ. (North is more driven by outside introductions, whereas South has more community transmission). But while the authors may be familiar with the differing demographics of these two regions, the reader has to make it all the way to Figure 6 (page 22) to get a picture of how these two regions differ socioeconomically, in % essential workers, etc. It would be helpful to move Figure 6 to the beginning of the paper and include some text in the introduction. Can the authors provide any clarity on why there seems to be such a stark socioeconomic line in the middle of King County? Why do adjacent zip codes in the middle of the county differ so much in socioeconomic measures? Could you also provide a map with population density in Figure 6? Possibly vaccination rates?

2. This study examines transmission in the context of non-pharmaceutical interventions like stay at home orders and closing bars. What about closing schools?

3. I could not find any MCC trees or ML trees in the GitHub repository. Figure 2 is missing background data so the King County data is more visible but there are some surprisingly large King County clusters, given how fluidly SARS-CoV-2 mixes within and between states. Are these monophyletic? And the very long branches (appears to be delta? some labeling of the different variants in the tree would help), are these chronically infected people? One of the branches spans >6 months. What % of transmission chains (maybe with at least 10 sequences) are only found in one region (north or south) versus both?

4. In your models, did you use the same parameters throughout, or change them as new variants arrived?

5. It would be helpful in all the figures to label the variants in the different waves (e.g., Fig 3). Most readers probably know them, but when people this read this paper 5-10 years from now they may forget.

6. Could your discussion include some more concrete examples of how this kind of molecular epidemiology could, in retrospect, change how policy makers acted during the pandemic? Based on these findings, could they have targeted interventions differently or more strategically? Is one takeaway that a region's COVID control is only as good as its weakest link, because any pocket where cases are high will quickly seed other areas? What is your main takeaway?

PLOS authors have the option to publish the peer review history of their article (what does this mean?). If published, this will include your full peer review and any attached files.

Reviewer #1: No

Reviewer #2: No

Figure Files:

Data Requirements:

Reproducibility:

References:

---

## [Editor Report · Decision Letter 1]

12 Mar 2024

Dear Mr. Paredes,

We are pleased to inform you that your manuscript 'Local-Scale phylodynamics reveal differential community impact of SARS-CoV-2 in metropolitan US county' has been provisionally accepted for publication in PLOS Pathogens.

Best regards,

Sonja M. Best, Ph.D.

Section Editor

PLOS Pathogens

Sonja Best

Section Editor

PLOS Pathogens

Michael Malim

Editor-in-Chief

PLOS Pathogens

orcid.org/0000-0002-7699-2064
---

## [Editor Report · Acceptance letter]

21 Mar 2024

Dear Mr. Paredes,

We are delighted to inform you that your manuscript, "Local-Scale phylodynamics reveal differential community impact of SARS-CoV-2 in a metropolitan US county," has been formally accepted for publication in PLOS Pathogens.

Best regards,

Michael Malim

Editor-in-Chief

PLOS Pathogens

orcid.org/0000-0002-7699-2064